# Single-gold etching at the hypercarbon atom of *C*-centred hexagold(I) clusters protected by chiral *N*-heterocyclic carbenes

Xiao-Li Pei [1,3], Pei Zhao [2], Hitoshi Ube [1], Zhen Lei [1,4], Masahiro Ehara [2] & Mitsuhiko Shionoya [1,3]

Chemical etching of nano-sized metal clusters at the atomic level has a high potential for creating metal number-specific structures and functions that are difficult to achieve with bottom-up synthesis methods. In particular, precisely etching metal atoms one by one from nonmetallic element-centred metal clusters and elucidating the relationship between their well-defined structures, and chemical and physical properties will facilitate future materials design for metal clusters. Here we report the single-gold etching at a hypercarbon centre in gold(I) clusters. Specifically, *C*-centred hexagold(I) clusters protected by chiral *N*-heterocyclic carbenes are etched with bisphosphine to yield *C*-centred pentagold(I) (**CAu$^I_5$**) clusters. The **CAu$^I_5$** clusters exhibit an unusually large bathochromic shift in luminescence, which is reproduced theoretically. The etching mechanism is experimentally and theoretically suggested to be a tandem dissociation-association-elimination pathway. Furthermore, the vacant site of the central carbon of the **CAu$^I_5$** cluster can accommodate AuCl, allowing for post-functionalisation of the *C*-centred gold(I) clusters.

Etching is a top-down method to downsize the structures at the atomic level and modify the chemical and physical properties of a wide range of nanomaterials such as nanocrystals and colloidal nanoparticles[1–5], and nanoclusters[6–11] for a variety of applications. For example, chemical etching methods that involve ligand engineering have made great advances, such as thiolate etching of phosphine- or thiolate-protected nanogold clusters[6–8], reverse etching of thiolated-Au$_{25}$ with phosphine (exchange of strong donor ligands for weaker donor ligands)[9], and phosphine exchanged by *N*-heterocyclic carbene (NHC) to give NHC-containing Au$_{11}$[10,11]. Rapid developments in X-ray crystallography have revealed that chemical etching alters the nanocluster structures of the metal core[12] and ligand surface[13–17]. However, the control and understanding of chemical etching at the atomic level have only just begun[18,19]. Recently, Cao et al. used real-time electrospray ionisation mass spectrometry to reveal degradation and anomalous

recombination processes in the chemical etching of Au$_{25}$ nanoclusters[19]. It also remains controversial whether the ligand-exchange mechanisms in nanogold regions containing Au$^I$ and Au$^0$ atoms is S$_N$2-like bimolecular nucleophilic substitution or S$_N$1-like type unimolecular nucleophilic substitution[20–22]. Despite the promise of chemical etching as a general technique to downsize metal clusters at the atomic level, little attention has been paid to ligand-protected Au$^I$ clusters.

Among the Au$^I$ cluster family, clusters radially coordinated to main-group elements[23–29] such as *O*-centre[23], *N*-centre[24,25], *C*-centre[26], and *S*-centre[27–29] are attractive due to the polyhedral structures similar to nanogold clusters, Au$^I$···Au$^I$ interactions[30–34] and structure-dependent photophysical properties. In particular, the hypercoordinated carbon (hypercarbon)[35]-centred hexagold(I) (**CAu$^I_6$**) cluster [C(Au$^I$-L)$_6$]X$_2$ (L = ligand; X = counterion)[26] that bridges nano-sized

[1]Department of Chemistry, Graduate School of Science, The University of Tokyo, Tokyo 113-0033, Japan. [2]Research Centre for Computational Science, Institute for Molecular Science and SOKENDAI, Myodaiji, Okazaki, Aichi 444-8585, Japan. [3]Present address: Research Institute for Science and Technology, Tokyo University of Science, 2641 Yamazaki, Noda, Chiba 278-8510, Japan. [4]Present address: Fujian Provincial Key Laboratory of Advanced Inorganic Oxygenated Materials, College of Chemistry, Fuzhou University, Fuzhou 350108, P. R. China. ✉e-mail: ehara@ims.ac.jp; shionoya@chem.s.u-tokyo.ac.jp

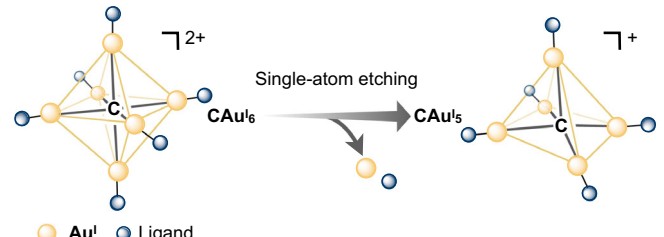

**Fig. 1 | The schematic illustration of etching ligand-protected *C*-centred gold(I) cluster.** Etching-induced elimination of [LAu$^I$] at the hypercarbon centre of the ligand-protected *C*-centred hexagold(I) (**CAu$^I_6$**) cluster results in the formation of the ligand-protected *C*-centred pentagold(I) (**CAu$^I_5$**) cluster, wherein the ligand can be an *N*-heterocyclic carbene or phosphine.

metal clusters and organic molecules[36,37] has attracted attention and significant advances have been made in this area. Many of these **CAu$^I_6$** clusters exhibit structure-specific luminescence[38–44] and can be used as bio-labels[39,44] by using highly bottom-up designable shell ligands based on phosphines[38–40] and NHCs[41–45]. However, there is only one isolated example of a *C*-centred pentagold(I) cluster [C(Au$^I$-L)$_5$]X (L = triphenylphosphine, TPP) produced by the bottom-up synthesis of aurating CH$_2$[B(OCH$_3$)$_2$]$_2$ with a gold(I) complex, and if the reaction time is extended, the TPP-protected **CAu$^I_6$** cluster is the main product[46]. Thus, it is still difficult to control the number of Au$^I$ atoms bound to the hypercarbon centre. In particular, from the viewpoint of the effects of reducing the number of Au$^I$ atoms on photophysical properties and reactivity, the development of a highly generalised single-gold etching method for gold(I) clusters is an important research topic.

Here, we discovered that a chiral NHC-protected *C*-centred hexagold(I) cluster can be etched with a bisphosphine ligand to generate a chiral NHC-protected **CAu$^I_5$** cluster. This was achieved by controlling the number of gold atoms centred at the hypercarbon at the atomic level (Fig. 1). Furthermore, this etching method is also useful for the synthesis of TPP-protected **CAu$^I_5$** analogues. In general, smaller gold clusters show more blue-shifted emission than larger clusters, but ligand-protected **CAu$^I_5$** clusters show unusually red-shifted signals in both absorption and emission spectra compared to the **CAu$^I_6$** counterparts, which was rationalised by theoretical calculations. Further experimental and theoretical studies suggest that a tandem dissociation-association-elimination pathway is involved in the etching mechanism. The NHC-protected **CAu$^I_6$** clusters are generally chemically stable, and the active site had to be placed on the hypercarbon to confer reactivity. In fact, **CAu$^I_5$** synthesised by this method was highly reactive with AuCl, producing a **CAu$^I_6$** cluster with a different ligand. Thus, the chemical etching method is expected to be a way not only to reduce the size of metal ion clusters and significantly change their electronic structure, but also to asymmetrise the metal ion cluster structure and provide active sites.

## Results

### Synthesis of CAu$^I_5$ by etching CAu$^I_6$ with bisphosphine

We previously reported enantiopure NHC-protected asymmetrically twisted **CAu$^I_6$** clusters: [(C)(Au$^I$-*SS*-NHC)$_6$](BF$_4$)$_2$ and [(C)(Au$^I$-*RR*-NHC)$_6$](BF$_4$)$_2$ ( = *SS*- and *RR*-**1$^{NHC}$**, *SS*-NHC = *N,N′*-bis[(*S*)-α-methylbenzyl]-benzimidazol-2-ylidene, *RR*-NHC = *N,N′*-bis[(*R*)-α-methylbenzyl]-benzimidazol-2-ylidene)[43]. Although NHCs are generally thought to bind strongly to coinage metal[47,48], we investigated whether etching occurs when bisphosphine is added to NHC-protected **CAu$^I_6$** clusters. For example, when 2.5 equiv. of *cis*-1,2-bis(diphenylphosphino)ethene (*cis*-depe) were added to a dichloromethane solution of *SS*-**1$^{NHC}$**, the original pale-yellow solution immediately turned orange. Subsequent crystallisation from diethyl ether/dichloromethane at 4 °C gave orange-red blocky crystals of [(C)(Au$^I$-*SS*-NHC)$_5$](BF$_4$) (*SS*-**2$^{NHC}$**) in 80%

yield on a hypercarbon basis. Its enantiomer [(C)(Au$^I$-*RR*-NHC)$_5$](BF$_4$) (*RR*-**2$^{NHC}$**) was also synthesised. Another product [(*cis*-depe)$_2$Au$^I$](BF$_4$) (**3**) was obtained as colourless blocky crystals by prolonged recrystallisation. They were characterised by ESI-MS spectrometry, NMR spectroscopy and elemental analysis (Suppl. Figs. 2–12). The ESI-MS spectrum found the **CAu$^I_5$** cluster *SS*-**2$^{NHC}$** at *m/z* 2628.85 corresponding to [(C)(Au$^I$-*SS*-NHC)$_5$]$^+$ (calcd. 2628.73). In the $^1$H NMR spectrum of the **CAu$^I_5$** cluster *SS*-**2$^{NHC}$** in *d$_6$*-acetone, the signals attributed to the NHC ligand showed a significant downshift compared to those of the **CAu$^I_6$** cluster *SS*-**1$^{NHC}$** (Suppl. Fig. 6). This is attributed to the magnetic environment, which is deshielded from the shell ligands with less steric hindrance. Similarly in the $^{13}$C NMR spectra, the signal at 198.8 ppm attributed to the NHC carbon-donors of *SS*-**2$^{NHC}$** was downshifted from those of *SS*-**1$^{NHC}$** at 190.0 ppm (Suppl. Fig. 7), suggesting a marked influence on resonances from different gold nuclearities.

Moreover, a phosphine analogue [(C)(Au$^I$-TPP)$_5$](BF$_4$) (**2$^{TPP}$**) was also obtained in 90% yield by etching [(C)(Au$^I$-TPP)$_6$](BF$_4$)$_2$ (**1$^{TPP}$**)[26] instead. The $^{31}$P NMR spectrum of **2$^{TPP}$** in *d$_6$*-acetone showed a singlet signal at 32.6 ppm, which was downshifted from that of **1$^{TPP}$** (Suppl. Fig. 8)[46]. Furthermore, etching *SS*-**1$^{NHC}$** and **1$^{TPP}$** with excess *cis*-depe (50 equiv.) yielded corresponding pentagold(I) clusters in both cases, with no detectable smaller gold species such as tetragold(I) or trigold(I) clusters. Etching *SS*-**1$^{NHC}$** and **1$^{TPP}$** with 1,2-bis(diphenylphosphino)benzene also yielded the corresponding **CAu$^I_5$** clusters. These data indicate that etching the **CAu$^I_6$** cluster with *cis*-depe provides high selectivity for the **CAu$^I_5$** cluster.

### Single-crystal structures and computational bonding analysis

The single-crystal X-ray diffraction (SCXRD) determined structures in Fig. 2 show the overall structure of *SS*- and *RR*-**2$^{NHC}$** including a hypercarbon, five gold(I) ions, five ligands and a BF$_4^-$ counterion. *SS*- and *RR*-**2$^{NHC}$** are crystallised in the *I*4 space group with low flack parameters of 0.010(9) and –0.027(11), respectively. Their flack parameters are very low (Suppl. Table 1), suggesting that optically pure molecules are packed. Take the example of *SS*-**2$^{NHC}$** as shown in Table 1, the Au$^I$···Au$^I$ distances (2.8667(10)–3.3141(15) Å) and the C$_{NHC}$–Au$^I$ bonds (2.03(2)–2.08(4) Å) are similar to those in *SS*-**1$^{NHC}$**, but the C$_{centre}$–Au$^I$ bonds (2.03(2)–2.075(8) Å) of *SS*-**2$^{NHC}$** are slightly shorter than those of *SS*-**1$^{NHC}$** (2.100(14)–2.126(12) Å)[43], suggesting that the endohedral five C$_{centre}$–Au$^I$ bonds in the **CAu$^I_5$** cluster are more favourable. The [(C)(Au$^I$-L)$_5$]$^+$ cation in *SS*-**2$^{NHC}$** can be regarded as eliminating one [LAu$^I$] moiety from the [(C)(Au$^I$-L)$_6$]$^{2+}$ cation in *SS*-**1$^{NHC}$**. As a result, the NHC ligands of *SS*-**2$^{NHC}$** rearrange themselves to minimise mutual steric hindrance (in Fig. 2b, three grey-coloured benzimidazolylidene moieties on the same plane and two orange-coloured benzimidazolylidene moieties on two planes with a 63° crossing angle), forming a monocationic **CAu$^I_5$** cluster with *C$_2$*-symmetry (Suppl. Fig. 15). It should be noted that the hypercarbon of *SS*-**2$^{NHC}$** is close to the centroid of the four gold atoms at the bottom of the square pyramid (0.46(3) Å), which could be an important coordinating site for post-functionalisation (*vide infra*). Meanwhile, the surface vacancy found in this *SS*-**2$^{NHC}$** molecule is well shielded in its packing structure by intermolecular interactions with a ligand on the gold(I) at the apex of another cluster molecule (Suppl. Fig. 16), thus maintaining high chemical stability in the solid state. In contrast, the phosphine-protected analogue **2$^{TPP}$** crystallised in the *P*2$_1$/*n* space group (Fig. 2c)[42] and exhibited Au$^I$···Au$^I$ interactions (2.85528(18)–3.21332(19) Å) and P–Au$^I$ bonds (2.2546(8)–2.2735(8) Å). The C$_{centre}$–Au$^I$ bonds (2.064(3)–2.082(3) Å) of **2$^{TPP}$** are slightly shorter than those of **CAu$^I_6$** counterpart (average 2.12 Å)[26], similar to the shorter C$_{centre}$–Au$^I$ bonds of *SS*-**2$^{NHC}$** compared to *SS*-**1$^{NHC}$**.

Moreover, the bonding characters of **CAu$^I_n$** (*n* = 5, 6) clusters were computationally studied based on the above crystallography data, and the bond distances of the crystal structures were well reproduced by the density functional theory (DFT) calculations. The calculated

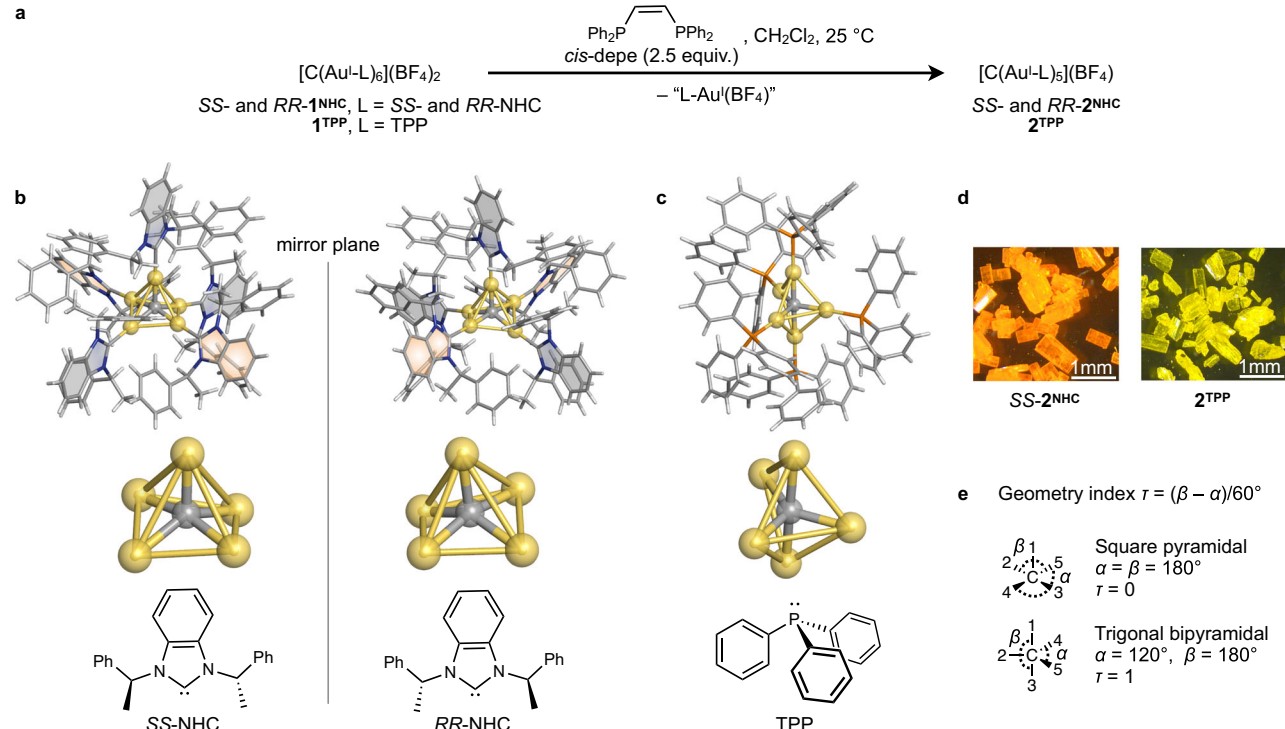

**Fig. 2 | Synthesis and characterisation of *C*-centred pentagold(I) clusters.**
**a** Etching syntheses of **CAu$^I_6$** clusters[26,43] to **CAu$^I_5$** clusters using *cis*-1,2-bis(diphe-nylphosphino)ethene (*cis*-depe) (Suppl. Fig. 1). **b** Single-crystal X-ray diffraction (SCXRD) structures of the cations [(C)(Au$^I$-L)$_5$]$^+$ (L = *SS*- and *RR*-NHC) and the **CAu$^I_5$** cores of *SS*- and *RR*-**2$^{NHC}$** with optically active NHC ligands. **c** SCXRD structures of the cation [(C)(Au$^I$-TPP)$_5$]$^+$ [46] and the core of **2$^{TPP}$** with TPP. **d** Microscopy images of crystals of *SS*-**2$^{NHC}$** (orange crystals) and **2$^{TPP}$** (yellow crystals) under ambient light.

**e** Explanation and examples of geometry index $\tau$ values[51], which is used for showing the geometric differences of **CAu$^I_5$** cores with NHCs and TPP (Table 1). The $\tau$ value represents the geometric difference between the regular square pyramidal ($\tau = 0$) and trigonal bipyramidal ($\tau = 1$), $\tau = (\beta - \alpha)/60°$, where $\alpha$ and $\beta$ are the two largest basal angles. Colour code: Au, yellow; C, grey; N, blue; P, orange; H, white. BF$_4^-$ counterions and solvent molecules are omitted for clarity.

C$_{centre}$−Au$^I$ and Au$^I$···Au$^I$ distances of **CAu$^I_n$** ($n$ = 5, 6) cores as well as Wiberg bond orders (WBO) suggested very interesting structural dependencies (Suppl. Table 3). The C$_{centre}$−Au$^I$ and Au$^I$···Au$^I$ distances of the $N,N'$-diisopropylimidazolidene (I*i*Pr)-protected **CAu$^I_6$** cluster[41] are 2.19 and 3.10 Å, respectively. Their bond orders are 0.41 and 0.16, respectively, indicating that the C$_{centre}$−Au$^I$ bond is stronger than each Au$^I$···Au$^I$ interaction. In the **CAu$^I_6$** cores of *SS*-**1$^{NHC}$** and **1$^{TPP}$**, the binding characteristics obtained are largely unchanged. Of note, the C$_{centre}$−Au$^I$ bonding in the **CAu$^I_5$** cores of *SS*-**2$^{NHC}$** and **2$^{TPP}$** is slightly stronger compared to the corresponding **CAu$^I_6$** clusters. This is demonstrated by the shorter bond lengths (2.09–2.16 Å) and larger WBO values (0.50–0.57). Regarding the aurophilic interactions in the **CAu$^I_5$** cores, the Au$^I$···Au$^I$ distances and bond orders are nearly the same as those of **CAu$^I_6$** cores. Therefore, missing one Au atom in the **CAu$^I_5$** cores may result in stronger C$_{centre}$−Au$^I$ bonds, which may be important for stabilising the **CAu$^I_5$** clusters. Moreover, the orbital interactions of the **CAu$^I_6$** cluster were previously discussed in detail[49,50]: the *SS*-**1$^{NHC}$** and

*SS*-**2$^{NHC}$** clusters have [CAu$_6$]$^{2+}$ and [CAu$_5$]$^+$ cores, respectively, and the C−Au$^I$ bond orders exhibit an unusual C−Au$^I$ bond hypervalence.

In addition, to better understand the geometric differences of **CAu$^I_5$** cores with NHCs and TPP, we introduced the index parameter $\tau$ (Fig. 2e, Table 1), wherein $\tau$ is 0 for perfect square pyramidal and 1 for perfect trigonal bipyramidal[51]. This evaluation method suggests that the **CAu$^I_5$** cores of *SS*- and *RR*-**2$^{NHC}$** have a distorted square pyramidal geometry ($\tau = 0.32$), while the **CAu$^I_5$** core of **2$^{TPP}$** is much closer to a trigonal bipyramidal geometry ($\tau = 0.68$), indicating that NHCs and TPP exert different ligand effects on the **CAu$^I_5$** core.

**Absorption, emission profiles, and theoretical calculations**
The **CAu$^I_6$** clusters are known to be efficient emitters with intriguing structure-dependent properties[38–45], while the photophysical properties of the **CAu$^I_5$** clusters remain unknown. In general, reducing the metal core size is known to induce a blue shift in absorption and emission[52]. However, the UV-vis spectra of *SS*-**2$^{NHC}$** and **2$^{TPP}$** in dichloromethane, in contrast, showed their maximum absorption wavelengths at 420 nm and 382 nm, respectively, and were significantly more red-shifted than those of **CAu$^I_6$** clusters, *SS*-**1$^{NHC}$** ($\lambda^{max}$ = 373 nm) and **1$^{TPP}$** ($\lambda^{max}$ = 365 nm) (Suppl. Fig. 17). Similarly, photoluminescence of the **CAu$^I_5$** clusters showed a bathochromic shift signal in contrast to the **CAu$^I_6$** clusters (Fig. 3a, Suppl. Fig. 18). The solid-state *SS*-**2$^{NHC}$** exhibited orange-red emission ($\lambda_{em}^{max}$ = 676 nm), which is 151 nm more red-shift than *SS*-**1$^{NHC}$**. The acetone solution of *SS*-**2$^{NHC}$** was also red-emissive, with no apparent solvation effects (Suppl. Fig. 19). On the other hand, the emission of **2$^{TPP}$** in the solid state is 59 nm more red-shifted than **1$^{TPP}$** and emits yellow at 365 nm excitation ($\lambda_{em}^{max}$ = 584 nm). Neither **1$^{TPP}$** nor **2$^{TPP}$** emits light in solution. This is because the terminal coordination of phosphine to the gold(I) atom

**Table 1 | Selected structural parameters of *SS*2$^{NHC}$ and 2$^{TPP}$**

|  | *SS*-**2$^{NHC}$** | **2$^{TPP}$** |
|---|---|---|
| Au$^I$···Au$^I$ (Å) | 2.8667(10)–3.3141(15) | 2.85528(18)–3.21332(19) |
| C$_{centre}$–Au$^I$ (Å) | 2.03(2)–2.075(8) | 2.064(3)–2.082(3) |
| C$_{NHC}$–Au$^I$ (Å) | 2.03(2)–2.08(4) | / |
| P–Au$^I$ (Å) | / | 2.2546(8)–2.2735(8) |
| $\tau$ | 0.32 | 0.68 |

Bond distances (Å) of Au$^I$···Au$^I$, C$_{centre}$–Au$^I$, C$_{NHC}$–Au$^I$ and P–Au$^I$, and geometry index $\tau$ values. The statistical significance of the errors for the bond distances is derived from the precision of the SCXRD data (Suppl. Table 1).

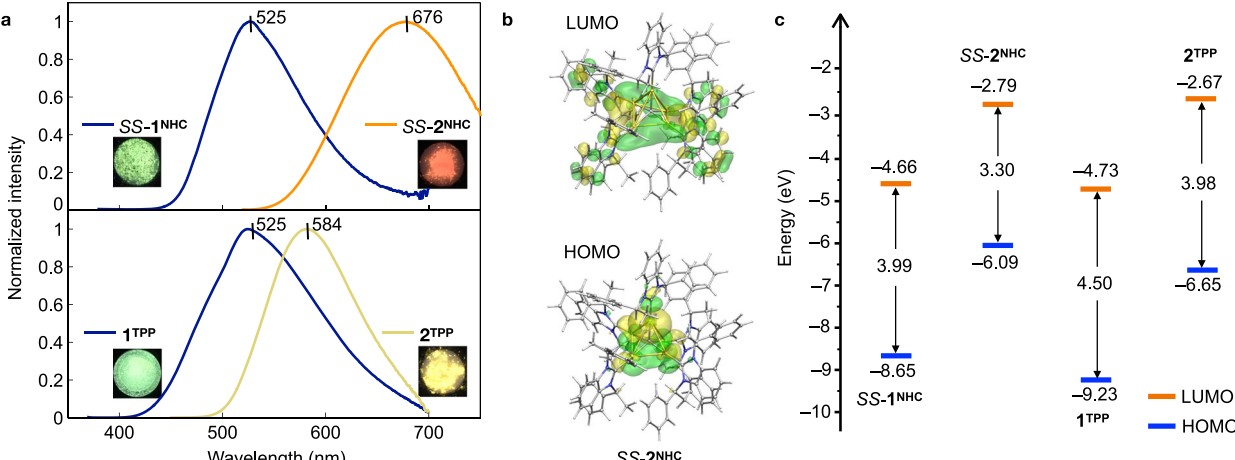

**Fig. 3 | Photoluminescence of *C*-centred gold(I) clusters in the solid state and the theoretical study. a** Emission spectra: *SS*-**1**$^{NHC}$ (blue line, excited by 266 nm), *SS*-**2**$^{NHC}$ (orange line, excited by 510 nm), **1**$^{TPP}$ (blue line, excited by 356 nm), **2**$^{TPP}$ (yellow line, excited by 365 nm), insets: photographs (size: 12 mm ×12 mm) of crystals under 365 nm UV-light irradiation. **b** The highest occupied molecular

orbital (HOMO) and the lowest unoccupied molecular orbital (LUMO) distributions of *SS*-**2**$^{NHC}$. **c** HOMO−LUMO gap values based on theoretical calculations: *SS*-**1**$^{NHC}$ (3.99 eV), *SS*-**2**$^{NHC}$ (3.30 eV), **1**$^{TPP}$ (4.50 eV), and **2**$^{TPP}$ (3.98 eV). Source data are provided as a Source Data file.

may facilitate nonradiative relaxation pathways[38]. Comparing the absolute quantum yields (*Φ*), *SS*-**2**$^{NHC}$ in the solid state showed the strongest emission. Here, the *Φ* values of *SS*-**2**$^{NHC}$, *SS*-**1**$^{NHC}$, **2**$^{TPP}$, and **1**$^{TPP}$ were 0.61, 0.02, 0.29, and 0.19, respectively (Suppl. Fig. 20). They exhibit microsecond-level lifetimes (Suppl. Fig. 21), suggesting phosphorescence properties in the solid state.

To gain insight into the optical properties specific to their electronic structures, we performed DFT and time-dependent (TD)-DFT calculations based on the SCXRD structures (Suppl. Figs. 22–27). The UV-vis spectrum of *SS*-**2**$^{NHC}$ calculated by TD-DFT (Suppl. Fig. 22) reproduces well the maximum band in the range from 393 to 415 nm, which is mainly due to the transitions of HOMO-1 → LUMO, HOMO-2 → LUMO and HOMO → LUMO + 1 (Suppl. Table 4). Orbital composition analysis by Mulliken partition (Suppl. Table 6) reveals that the occupied orbitals of HOMO-*n* (*n* = 0–2) in *SS*-**2**$^{NHC}$ are mainly derived from gold(I) ions (52.6–55.4%) with an increased contribution from the hypercarbon (26.9–32.5%) compared with that in *SS*-**1**$^{NHC}$ [43]. The unoccupied orbitals of LUMO+*n* (*n* = 0, 1) in *SS*-**2**$^{NHC}$ are localised mainly at NHCs (64.7–74.8%), with small fractions of gold(I) ions (25.0–35.2%). Thus, the dominant metal-to-ligand charge-transfer (MLCT) mixed with a slight metal-centred (MC) charge-transfer was responsible for the low-energy absorption bands[42]. On the other hand, the compositions of the frontier orbitals of the TPP-protected **CAu**$^I_n$ (*n* = 5, 6) clusters are also comparable (Suppl. Tables 7, 8, Suppl. Figs. 26, 27). These theoretical results suggest that the *C*-centred gold(I) core and ligands are essentially enrolled in their electronic structures, and thus different optical features can be explained by altering the gold nuclearity and ligands primarily via the MLCT transition[42].

Importantly, the HOMO−LUMO gaps calculated for these **CAu**$^I_n$ (*n* = 5, 6) clusters (Fig. 3c) show that the gaps of **CAu**$^I_5$ clusters are clearly smaller than those of the **CAu**$^I_6$ clusters: 3.30 eV (*SS*-**2**$^{NHC}$) < 3.99 eV (*SS*-**1**$^{NHC}$), 3.98 eV (**2**$^{TPP}$) < 4.50 eV (**1**$^{TPP}$). This is in good agreement with the fact that the absorptions of **CAu**$^I_5$ clusters are more red-shifted than those of the **CAu**$^I_6$ clusters. The calculated phosphorescence energies well reproduce the smaller phosphorescence energy of *SS*-**2**$^{NHC}$ (1.83 eV) than that of *SS*-**1**$^{NHC}$ (2.36 eV) (Suppl. Table 9) observed in experiments, which confirms that the emission of *SS*-**2**$^{NHC}$ is red-shifted and reveals that the *C*-centred gold(I) clusters have a pronounced size-dependent effect on the photoluminescence properties.

## Probing the atomic-level etching process

To understand the mechanism of this efficient etching method to reshape *C*-centred gold(I) clusters, we investigated this process using UV-vis absorption spectroscopy, NMR spectroscopy and ESI-MS spectrometry. First, the UV-vis spectra of *SS*-**1**$^{NHC}$ etched with *cis*-depe showed a rapid change (Fig. 4a,b). When *cis*-depe was added to a dichloromethane solution of *SS*-**1**$^{NHC}$ (*c* = 5 × 10$^{-5}$ M, 293 K), its characteristic peak ($\lambda^{max}$ = 345 nm) gradually decreased and new peaks appeared at 385 nm and 421 nm derived from *SS*-**2**$^{NHC}$. Accordingly, the original pale-yellow solution turned bright yellow. In contrast, the time-course UV-vis spectra of **1**$^{TPP}$ etched with *cis*-depe (Suppl. Fig. 28) showed that the peak at 382 nm appeared even more instantaneously for **2**$^{TPP}$. The colourless solution turned yellow within 5 s, a much faster change than etching *SS*-**1**$^{NHC}$. This can be reasonably explained by the weaker binding of triphenylphosphine to gold than NHC[48], consistent with the shorter C$_{NHC}$−Au$^I$ bonds than P−Au$^I$ bonds in the single-crystal structures.

Next, the $^1$H NMR spectra of *SS*-**1**$^{NHC}$ etched with *cis*-depe in $d_6$-acetone were measured over time (Fig. 4c–e). The results showed that *SS*-**2**$^{NHC}$ was formed in 98 % yield 0.5 h after the addition of *cis*-depe (1,3,5-trimethoxybenzene as the internal standard), with the detection of another product [(*cis*-depe)$_2$Au$^I$](BF$_4$) (**3**) (Suppl. Fig. 29). On the other hand, when **1**$^{TPP}$ was etched with *cis*-depe, the time-course of the $^1$H NMR spectra in $d_6$-acetone showed that **2**$^{TPP}$ was formed in 88% yield after 18 min (Suppl. Fig. 30). The $^{31}$P NMR spectrum in $d_6$-acetone after the reaction showed signals for complex **3** ($\delta$ 21.3 ppm) and free triphenylphosphine ($\delta$ −4.2 ppm). Furthermore, we turned to ESI-MS spectrometry to obtain more molecular information. As a result, a signal corresponding to *SS*-**2**$^{NHC}$ was observed at *m/z* 2628.71 (calcd. 2628.73 for [(C)(Au$^I$-*SS*-NHC)$_5$]$^+$) immediately after adding *cis*-depe, and the very weak signals of two intermediates **Int1**$^{NHC}$ and **Int2**$^{NHC}$ were found (Suppl. Fig. 31). Interestingly, in the process of etching **1**$^{TPP}$, two di-cationic mass peaks, **Int1**$^{TPP}$ [(C)(Au$^I$-TPP)$_6$]$^{2+}$ (*m/z* found 1252.08, calcd. 1252.12) and **Int2**$^{TPP}$ ([(C)(Au$^I$-TPP)$_5$(Au$^I$-*cis*-depe)]$^{2+}$ (*m/z* found 1450.62, calcd. 1450.86) were detected (Suppl. Fig. 32). However, no information was available for the association adduct [(C)(Au$^I$-TPP)$_6$(*cis*-depe)]$^{2+}$ by binding *cis*-depe to **1**$^{TPP}$. This would suggest that the initial stage of etching phosphine-protected **CAu**$^I_6$ cluster with *cis*-depe is a dissociation process. Similarly, the formation of the association adduct [(C)(Au$^I$-*SS*-NHC)$_6$(*cis*-depe)]$^{2+}$ would be difficult due to the steric hindrance from NHC ligands in *SS*-**1**$^{NHC}$. Overall, as shown in

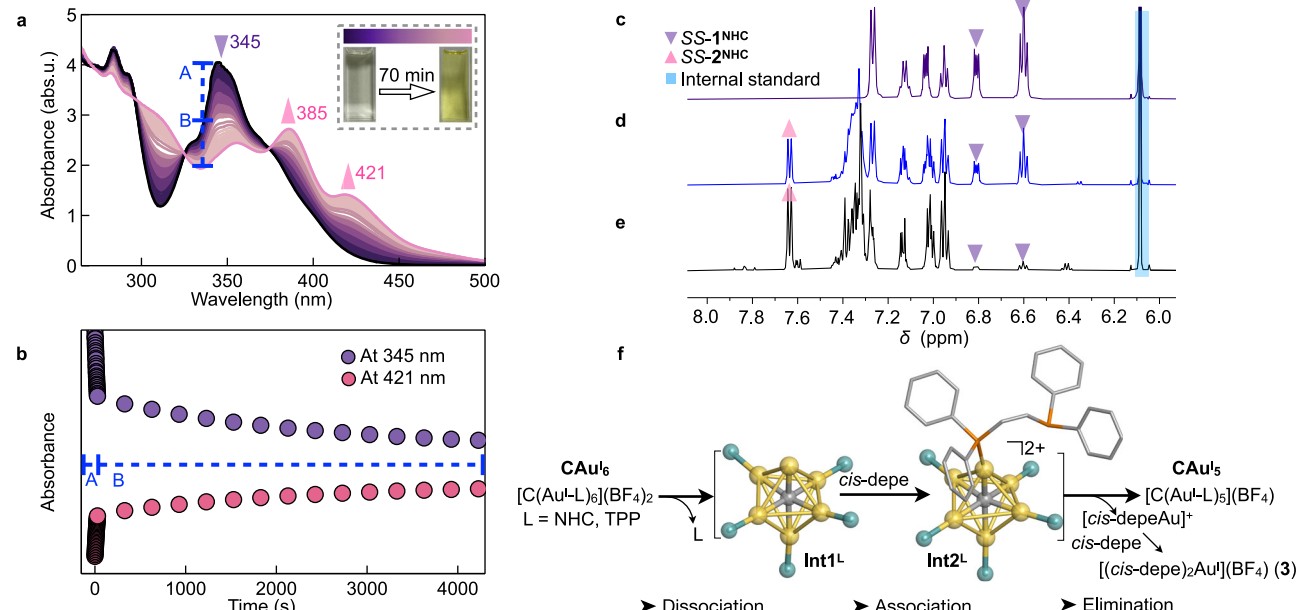

**Fig. 4 | Monitoring the etching process. a** Time-course UV-vis absorption spectra of the etching reaction of *SS*-**1**^NHC (*c* = 5 × 10^−5 M, 293 K) using *cis*-depe in dichloromethane: stage A, 1 s intervals; stage B, 5 min intervals; insets: photographs taken at the beginning of the reaction and after 70 min under ambient light, and a colour scale corresponding to the absorbance changes of the reaction. **b** The changes in absorbance at 345 nm and 421 nm as a function of time (corresponding to the spectra in **a**), suggesting that *SS*-**2**^NHC was rapidly formed. **c** ^1H NMR spectrum (*d_6*-acetone, 300 K), showing signals of *SS*-**1**^NHC (purple) and internal standard (IS, 1,3,5-trimethoxybenzene, blue-labelled). **d** Signals from *SS*-**2**^NHC (pink-labelled, 53% yield) measured immediately after adding *cis*-depe. **e** After 0.5 h, *SS*-**2**^NHC was formed in 98% yield. **f** A proposed etching mechanism with two intermediates **Int1**^L and **Int2**^L (L = NHC, TPP), with **Int1**^TPP and **Int2**^TPP detected by ESI-MS spectrometry (Suppl. Fig. 32). Colour code: Au, yellow; C, grey; L, cyan; P, orange. Source data are provided as a Source Data file.

Fig. 4f, the initial dissociation process similar to the ligand-exchange S_N1-like mechanism[22] generating the first intermediates **Int1**^L (L = TPP or NHC) would occur when etching the ligand-protected **CAu**^I_6 clusters. Subsequent association with *cis*-depe would form the second intermediates **Int2**^L (L = TPP or NHC), and then the elimination of Au^I with *cis*-depe finally produce the corresponding **CAu**^I_5 clusters.

To illustrate the proposed etching mechanism as shown in Fig. 4f, we computed the energy profiles of the etching process in dichloromethane (Suppl. Fig. 33). In the first dissociation stage, to break one of the six Au^I–L bonds from the original ligand-protected **CAu**^I_6 cluster, for example when L = TPP, **Int1**^TPP formed in relatively high energy (30.2 kcal mol^−1), suggesting the high stability of **1**^TPP. In the second association stage, when *cis*-depe coordinated to **Int1**^TPP, the resulting **Int2**^TPP was largely stabilised with a dramatically decreased energy of 10.9 kcal mol^−1. Third, followed by the elimination of [*cis*-depeAu]^+ to form a highly stable complex **3**, **2**^TPP was finally formed with an energy of 16.4 kcal mol^−1 via breaking the C_centre–Au^I bond and four Au^I⋯Au^I contacts of **Int2**^TPP. The energy profiles were similarly illustrated when L = NHC (Suppl. Fig. 33b). Therefore, the theoretical data supported the tandem dissociation-association-elimination pathway in this etching process.

## Stability study of CAu^I_5 clusters

*SS*-**2**^NHC and **2**^TPP in the solid state are stable for more than a year under ambient conditions, but they are reactive in solution, in contrast to the more stable **CAu**^I_6 clusters[43]. The changes over time of ^1H NMR spectra indicate that the solution of *SS*-**2**^NHC in *d_6*-acetone is stable for at least one week (Suppl. Fig. 34). On the other hand, once dissolved in CDCl_3, both *SS*-**2**^NHC and **2**^TPP partially reverted to the **CAu**^I_6 cluster (Suppl. Figs. 35, 36), and the original yellowish solutions gradually faded. These results suggest that the **CAu**^I_5 clusters are more reactive in solution than the corresponding **CAu**^I_6 clusters, which is consistent with the higher energies of the **CAu**^I_5 clusters in the calculated energy profiles described above.

## Reactivity of the CAu^I_5 cluster

The metal clusters with exposed surfaces are of growing interest[53]. The more open-spaced coordination site at the bottom of the distorted square pyramidal structures of *SS*- and *RR*-**2**^NHC were assumed to be the site where the sixth Au^I species is most accessible to the hypercarbon. Therefore, we added an acetone solution of (tht)Au^ICl (1 equiv.) to an acetone solution of *SS*-**2**^NHC at room temperature. The reaction was conducted in an ultrasonication bath for 20 min, and the crystallisation yielded a heteroleptic Cl-coordinated **CAu**^I_6 cluster [(C)(Au^I-*SS*-NHC)_5(Au^ICl)](BF_4) (*SS*-**4**^NHC) (Fig. 5, see characterisation data in Suppl. Figs. 37–41, Suppl. Table 10). Its enantiomer *RR*-**4**^NHC was similarly obtained. In the overall SCXRD structures of *SS*- and *RR*-**4**^NHC, in particular, the structures corresponding to the five NHC ligand parts of the **CAu**^I_5 clusters, *SS*- and *RR*-**2**^NHC, were found to be largely intact (Suppl. Fig. 37). The sixth gold(I) atom was coordinated to the Cl-anion with bond distances of 2.308(6) Å and 2.310(7) Å in *SS*- and *RR*-**4**^NHC, respectively. However, the introduction of AuCl into **2**^TPP was not successful, probably due to its low stability, and only the original **CAu**^I_6 cluster **1**^TPP was finally isolated. Given the NHC-protected monogold chloride complexes are extensively used as active catalysts[54–56], this cluster-based analogue Cl-coordinated **CAu**^I_6 cluster protected by the NHC ligands will be a milestone in the development of highly reactive hexagold(I) clusters.

In addition, the Cl-coordinated **CAu**^I_6 clusters *SS*- and *RR*-**4**^NHC restored very weak green emission in the solid state and no emission was observed in the solution at room temperature. In particular, the circular dichroism spectra of *SS*- and *RR*-**2**^NHC and *SS*- and *RR*-**4**^NHC in dichloromethane (Suppl. Fig. 42) showed similar chiroptical signals with the strongest signal at 250 nm contributed mainly by the chiral ligands, which can be explained by the similar arrangement of the ligand shell and symmetric metal cores. In a word, the etch-produced **CAu**^I_5 clusters (*SS*- and *RR*-**2**^NHC) and the post-functionalised Cl-coordinated **CAu**^I_6 clusters (*SS*- and *RR*-**4**^NHC) exhibit different properties from the original **CAu**^I_6 clusters (*SS*- and *RR*-**1**^NHC), thus revealing

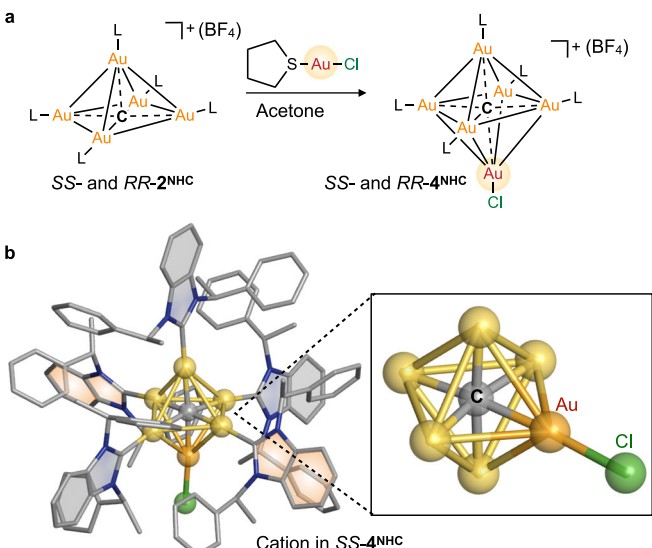

**Fig. 5 | Reactivity of the hypercarbon in CAu$^I_5$ clusters. a** Reaction of *SS*-(or *RR*-) **2**$^{NHC}$ with (tht)Au$^I$Cl (tht = tetrahydrothiophene) yielded Cl-coordinated **CAu$^I_6$** clusters: *SS*- and *RR*-**4**$^{NHC}$ [(C)(Au$^I$-L)$_5$(Au$^I$Cl)](BF$_4$) (L = *SS*- and *RR*-NHC). **b** The SCXRD structure of the [(C)(Au$^I$-*SS*-NHC)$_5$(Au$^I$Cl)]$^+$ cation and the Cl-coordinated **CAu$^I_6$** core in *SS*-**4**$^{NHC}$. The arrangement of three grey-coloured benzimidazolylidene moieties on the same plane and two orange-coloured benzimidazolylidene moieties on two planes in *SS*-**4**$^{NHC}$ is similar to that in *SS*-**2**$^{NHC}$ (Suppl. Fig. 37). Colour code: Au, yellow and orange; C, grey; N, blue; Cl, green. Hydrogen atoms, BF$_4^-$ counterion, and solvent molecules are omitted for clarity.

intriguing structure-property relationships by using etching as atomically precise "surgery" at the hypercarbon atom.

## Discussion

In summary, we have shown that etching of the NHC-protected **CAu$^I_6$** clusters allows size-selective synthesis of the corresponding **CAu$^I_5$** clusters. The peculiar red-shift signals in the absorption and emission of **CAu$^I_5$** clusters can be explained by theoretical calculations. A tandem dissociation-association-elimination pathway for the atomical-level etching was proposed based on experimental and theoretical studies. The envisaged coordination ability of the hypercarbon atom in **CAu$^I_5$** clusters was confirmed by adding Au$^I$Cl, leading to the novel heteroleptic Cl-coordinated **CAu$^I_6$** clusters. These results of the single-gold etching of the **CAu$^I_6$** clusters at the atomic level indicate a unique and highly generalised method using phosphine ligands for etching of NHC-protected gold clusters. This study not only elucidated the unusual photophysical properties of metal clusters containing fewer metal nuclei, but also provided opportunities to explore post-functionalisation and reactivities in surface-exposed metal ion clusters. This result shows that establishing a synthesis method using precision etching of **CAu$^I_n$** (*n* < 6) clusters is important for elucidating the chemical and physical properties and reactivity of unsymmetric clusters. Therefore, the chemical etching method is a way to reduce the size of metal ion clusters and will be developed to control the electronic structure, asymmetrisation of the metal ion cluster structure, catalytic reactions, and metal ion exchange.

## Methods
### NMR spectra
$^1$H, $^{13}$C and 2D NMR spectra were measured on a Bruker AVANCE III-500 (500 MHz) spectrometer. The residual solvent signal was used to calibrate the $^1$H (7.26 ppm), $^{13}$C NMR (77.16 ppm) measurements when CDCl$_3$ was used. The residual solvent signal was used to calibrate the $^1$H

(2.05 ppm), $^{13}$C NMR (29.84 ppm) measurements when $d_6$-acetone was used.

### ESI-MS analysis
ESI-TOF-MS data were measured on a Micromass LCT Premier XE mass spectrometer. Unless otherwise noted, the experimental conditions were as follows (ion mode, positive; capillary voltage, 2400 V; sample cone voltage, 30 V; desolvation temperature, 150 °C; source temperature, 80 °C).

### Elemental analysis
Elemental analyses (C, N, H) were conducted in the microanalytical laboratory, Department of Chemistry, School of Science, the University of Tokyo, using a Vario MICRO Cube elemental analyser with MgO added.

### Single-crystal X-ray diffraction analysis
X-ray crystallographic analysis was performed using a Rigaku XtaLAB PRO MM007DW PILATUS diffractometer with MoK$\alpha$ and CuK$\alpha$ radiation (93 K), and the obtained data were calculated using the Crystal Structure crystallographic software package. The refinement was performed using an OLEX2 software[57] with SHELXT[58]. All hydrogen atoms were geometrically placed and refined using the riding model.

### Photophysical analysis
The UV-vis absorption spectra were measured on a JASCO V-770 UV-vis spectrophotometer, wherein the temperature is set at 293 K unless otherwise mentioned. The emission and excitation spectra were measured using a Jasco FP-8300 fluorometer. The absolute quantum yields and lifetime measurements in the solid state were measured using Quantaurus-QY (Hamamatsu C9920-02G) and Hamamatsu C11367-02, respectively. CD spectra were measured on a JASCO J-820 circular dichroism spectrometer. The experimental conditions of CD analysis were as follows: bandwidth, 1 nm; response, 0.5 s; data acquisition interval, 0.5 nm; scanning rate, 100 nm min$^{-1}$.

### DFT and TD-DFT calculations
We applied the B3LYP functional[59] for geometry optimizations and TD-DFT calculations. The relativistic effective core potential LANL2DZ[60] was used for the Au atoms, and the basis set for the other atoms was 6-31G*[61]. Since MLCTs in the Au clusters etched in this study do not show long-distance charge transfer and essentially correspond to charge reorganisation, global hybrid functionals such as B3LYP adequately describe these electronic transitions. For simulating absorption spectra, 200 excited states were solved to cover the spectrum in the energy range up to about 220 nm was calculated in the velocity form. TD-DFT calculation was conducted with the polarizable continuum model (PCM) and the non-equilibrium linear response scheme[62], including the solvent effect of CH$_2$Cl$_2$. All calculations were conducted using the Gaussian 16 suite of programs[63]. The orbital composition was analysed using the Multiwfn program[64].

### Handling
All the syntheses were conducted under air unless otherwise mentioned.

### Chemical reagents
Unless otherwise noted, all the solvents were purchased from WAKO Pure Chemical Industries Ltd. and used without further purification. The >96% (NMR) pure *cis*-1,2-bis(diethylphosphino)ethene (*cis*-depe) was purchased from WAKO Pure Chemical Industries Ltd. and the >98% (GC) pure 1,2-bis(diethylphosphino)benzene was purchased from TCI Co., Ltd., and used without further purification. The starting materials of carbon-centred hexagold(I) clusters [(C)(Au$^I$-*SS*-NHC)$_6$]

$(BF_4)_2$ ($SS$-$\mathbf{1^{NHC}}$) [(C)(Au$^I$-$RR$-NHC)$_6$](BF$_4$)$_2$ ($RR$-$\mathbf{1^{NHC}}$)[43] and [(C)(Au$^I$-TPP)$_6$](BF$_4$)$_2$ ($\mathbf{1^{TPP}}$)[26] were synthesised according to the reported procedures.

## Synthesis of [(C)(Au$^I$-$SS$-NHC)$_5$](BF$_4$) ($SS$-$\mathbf{2^{NHC}}$) and [(C)(Au$^I$-$RR$-NHC)$_5$](BF$_4$) ($RR$-$\mathbf{2^{NHC}}$)

To the solution of $SS$-$\mathbf{1^{NHC}}$ [(C)(Au$^I$-$SS$-NHC)$_6$](BF$_4$)$_2$ (6.7 mg, 2 μmol) in CH$_2$Cl$_2$ (1 mL), a solution of 2.5 equiv. of $cis$-depe (5 μmol, 2.0 mg) in CH$_2$Cl$_2$ (1 mL) was added dropwise at room temperature. The original pale-yellow solution turned orange immediately. Next, the resulting reaction mixture was concentrated to 0.3 mL using an evaporator and then filtered into a tube through the cotton, finally layered with 3 mL Et$_2$O for slow diffusion and stored in a refrigerator at 4 °C. After one day, orange-red blocky crystals of $SS$-$\mathbf{2^{NHC}}$ [(C)(Au$^I$-$SS$-NHC)$_5$](BF$_4$) were formed and isolated (4.3 mg, 80% yield, based on the hypercarbon). The $RR$-$\mathbf{2^{NHC}}$ [(C)(Au$^I$-$RR$-NHC)$_6$](BF$_4$) (4.6 mg, 85% yield, based on the hypercarbon) is obtained similarly by using $RR$-$\mathbf{1^{NHC}}$ [(C)(Au$^I$-$RR$-NHC)$_6$](BF$_4$)$_2$ (6.7 mg, 2 μmol) as starting material. Anal. calcd. for [C$_{116}$H$_{110}$Au$_5$BF$_4$N$_{10}$](CH$_2$Cl$_2$)$_2$: C, 49.11; H, 3.98; N, 4.85. Found: C, 49.20; H, 4.23; N, 4.96. ESI-MS (CH$_2$Cl$_2$, positive): [$SS$-$\mathbf{2^{NHC}}$]$^+$ [C$_{116}$H$_{110}$N$_{10}$Au$_5$]$^+$, $m/z$ 2628.85 (calcd. 2628.73). ESI-MS (CH$_2$Cl$_2$, positive): [$RR$-$\mathbf{2^{NHC}}$]$^+$ [C$_{116}$H$_{110}$N$_{10}$Au$_5$]$^+$, $m/z$ 2628.92 (calcd. 2628.73). $^1$H NMR (500 MHz, 300 K, CDCl$_3$): $\delta$ 7.66–7.60 (m, 4H), 7.38 (q, $J$ = 7.4 Hz, 2H), 7.16–7.12 (m, 2H), 7.03–7.01 (m, 2H), 7.00 (d, $J$ = 1.3 Hz, 1H), 6.95 (t, $J$ = 7.4 Hz, 3H), 1.73 (d, $J$ = 7.3 Hz, 5H). $^{13}$C NMR (126 MHz, 300 K, CDCl$_3$): $\delta$ 206.1 (C$_{NHC}$), 198.8, 140. 1 , 133.2, 129.3, 128.6, 128.1, 123.8, 114.1, 59.0 (−CH−), 17.5 (−CH$_3$). In the $^{13}$C NMR spectrum, the signal of the hypercarbon atom was not detected even after a long-time accumulation.

## Synthesis of [(C)(Au$^I$-TPP)$_5$](BF$_4$) (2$^{TPP}$)

The synthesis of $\mathbf{2^{TPP}}$ is similar to that of $SS$-$\mathbf{2^{NHC}}$ by using $\mathbf{1^{TPP}}$ [(C)(Au$^I$-TPP)$_6$](BF$_4$)$_2$ as the starting material. After adding a solution of $cis$-depe (25 μmol, 9.9 mg, 2.5 equiv.) in CH$_2$Cl$_2$ (1 mL) to a solution of $\mathbf{1^{TPP}}$ (29.4 mg, 10 μmol) in CH$_2$Cl$_2$ (1 mL), the resulting mixture turned from colourless to yellow. Next, the resulting reaction mixture was concentrated to 0.3 mL using an evaporator, and then filtered into a tube through the cotton, finally layered with 3 mL Et$_2$O for slow diffusion and stored in a refrigerator at 4 °C. After several days, the yellow blocky crystals of $\mathbf{2^{TPP}}$ were isolated (22.3 mg, yield 93%, based on the hypercarbon). Anal. calcd. for [C$_{91}$H$_{75}$Au$_5$BF$_4$P$_5$]: C, 45.63; H, 3.16; N, 0. Found: C, 45.61; H, 3.24; N, 0.22. ESI-MS (CH$_2$Cl$_2$, positive): [$\mathbf{2^{TPP}}$]$^+$ [C$_{91}$H$_{75}$P$_5$Au$_6$]$^+$, $m/z$ 2307.21 (calcd. 2307.29). $^1$H NMR (500 MHz, 300 K, CDCl$_3$): $\delta$ 7.61–7.53 (m, 2H), 7.46–7.37 (m, 2H), 7.19 (td, $J$ = 7.9, 1.8 Hz, 3H). $^{31}$P NMR (202 MHz, 300 K, CDCl$_3$): $\delta$ 32.59 (s); $^{13}$C NMR (126 MHz, 300 K, CDCl$_3$): $\delta$ 134.9 (d, $J$ = 14.9 Hz), 133.0 (d, $J$ = 50.3 Hz), 131.9, 129.9 (d, $J$ = 11.4 Hz). In the $^{13}$C NMR spectrum, the signal of the hypercarbon atom was not detected even after a long-time accumulation.

## Synthesis of complex [($cis$-depe)$_2$Au$^I$](BF$_4$) (3)

In the above-mentioned synthesis of $SS$-$\mathbf{2^{NHC}}$, after isolating the desired crystals of $SS$-$\mathbf{2^{NHC}}$. The residue was used for recrystallisation, and several colourless crystals of [($cis$-depe)$_2$Au$^I$](BF$_4$) (3) (yield 50% by $^1$H NMR) were formed after one week. Similarly, after isolating crystals of $\mathbf{2^{TPP}}$ from its crystallisation tube, wherein the residue was recrystallised to give 3 (yield 99%, by $^1$H NMR). Its single crystal structure was determined by SCXRD (Suppl. Fig. 14). Its $^1$H NMR spectrum is consistent with literature[65].

## Synthesis of [(C)(Au$^I$-$SS$-NHC)$_5$(AuCl)](BF$_4$) ($SS$-$\mathbf{4^{NHC}}$) and [(C)(Au$^I$-$RR$-NHC)$_5$(AuCl)](BF$_4$) ($RR$-$\mathbf{4^{NHC}}$)

To a solution of $SS$-$\mathbf{2^{NHC}}$ (16 mg, 6 μmol) in 10 mL acetone, 1.2 equiv. of (tht)AuCl (2.3 mg, 7 μmol) in acetone (1 mL) was added dropwise (note: fast mixing caused decomposition and the formation of black precipitates) with continuous ultrasonic oscillation for 20 min. On completion, the resulting mixture was concentrated to 0.5 mL, and then

filtered into a tube through the cotton, finally layered with 3 mL Et$_2$O for slow diffusion and stored in a refrigerator at 4 °C. After several days, dark blocky crystals of [(C)(Au$^I$-$SS$-NHC)$_5$(AuCl)](BF$_4$) $SS$-$\mathbf{4^{NHC}}$ was isolated (10.0 mg, 57% yield, based on the hypercarbon). Anal. calcd. for [C$_{116}$H$_{110}$Au$_6$BClF$_4$N$_{10}$](CH$_3$COCH$_3$)(H$_2$O): C, 47.26; H, 3.93; N, 4.63. Found: C, 46.90; H, 4.34; N, 4.86. ESI-MS (CH$_2$Cl$_2$, positive): [$SS$-$\mathbf{4^{NHC}}$]$^+$ [C$_{116}$H$_{110}$N$_{10}$ClAu$_6$]$^+$, $m/z$ 2860.71 (calcd. 2860.66). ESI-MS (CH$_2$Cl$_2$, positive): [$RR$-$\mathbf{4^{NHC}}$]$^+$ [C$_{116}$H$_{110}$N$_{10}$ClAu$_6$]$^+$, $m/z$ 2860.70 (calcd. 2860.66). $^1$H NMR (500 MHz, 300 K, CDCl$_3$): $\delta$ 7.64 (d, $J$ = 7.6 Hz, 4H), 7.38 (q, $J$ = 7.3 Hz, 2H), 7.23–7.22 (m, 2H), 7.11–7.02 (m, 4H), 6.96 (t, $J$ = 7.6 Hz, 4H), 1.62 (d, $J$ = 7.3 Hz, 6H). $^{13}$C NMR (126 MHz, 300 K, CDCl$_3$): $\delta$ 189.93 (C$_{NHC}$), 139.8, 132.8, 129.4, 128.8, 128.0, 124.4, 114.6, 59.8 (−CH−), 17.3 (−CH$_3$). In the $^{13}$C NMR spectrum, the signal of the central carbon atom was not detected even after a long-time accumulation.

## Kinetic studies of the etching process

**Time-course experiments monitored by UV-vis spectroscopy.** Etching [(C)(Au$^I$-$SS$-NHC)$_6$](BF$_4$)$_2$ ($SS$-$\mathbf{1^{NHC}}$) with $cis$-depe. A solution of $SS$-$\mathbf{1^{NHC}}$ (5.0 ×10$^{-5}$ M) in CH$_2$Cl$_2$ was prepared by dissolving $SS$-$\mathbf{1^{NHC}}$ (1.7 mg, 0.5 μmol) in 10 mL CH$_2$Cl$_2$. We first measured this dichloromethane solution of $SS$-$\mathbf{1^{NHC}}$ (5.0 ×10$^{-5}$ M, 3 mL) by UV-vis spectroscopy at 293 K. Then, once 30 μL of $cis$-depe (2 equiv., 0.01 M) in CH$_2$Cl$_2$ was added (meanwhile, the solution was charged with a small magnetic stir and stirring at a rate of 60 rpm), the UV-vis spectra (Fig. 4a in the main text) of the resulting reaction were immediately measured at intervals of 1 s (stage A), and then at intervals of 5 min (stage B).

Etching [(C)(Au$^I$-TPP)$_6$](BF$_4$)$_2$ ($\mathbf{1^{TPP}}$) with $cis$-depe. A solution of $\mathbf{1^{TPP}}$ (5.0 ×10$^{-5}$ M) in CH$_2$Cl$_2$ was prepared by dissolving $\mathbf{1^{TPP}}$ (1.5 mg, 0.5 μmol) in 10 mL CH$_2$Cl$_2$. We first measured this dichloromethane solution of $\mathbf{1^{TPP}}$ (5.0 ×10$^{-5}$ M, 3 mL) by UV-vis spectroscopy at 293 K. Then, once 30 μL of $cis$-depe (2 equiv., 0.01 M) in CH$_2$Cl$_2$ was added (meanwhile, the solution was charged with a small magnetic stir and stirring at a rate of 60 rpm), the UV-vis spectra of the resulting reaction were immediately measured at intervals of 1 s (Suppl. Fig. 28).

**Time-course experiments monitored by $^1$H NMR spectroscopy.** Etching [(C)(Au$^I$-$SS$-NHC)$_6$](BF$_4$)$_2$ ($SS$-$\mathbf{1^{NHC}}$) with $cis$-depe. A solution of $SS$-$\mathbf{1^{NHC}}$ (2.0 ×10$^{-3}$ M) in $d_6$-acetone was prepared by dissolving $SS$-$\mathbf{1^{NHC}}$ (3.3 mg, 1 μmol) in $d_6$-acetone (0.5 mL) in the presence of an internal standard (IS, 1,3,5-trimethoxybenzene, 6 equiv., 6 μmol, 1.0 mg). We firstly measured the solution of $SS$-$\mathbf{1^{NHC}}$ in $d_6$-acetone by $^1$H NMR spectroscopy at 300 K. Then, once the solution of $cis$-depe (2.5 equiv., 3.3 ×10$^{-2}$ M, 75 μL) in $d_6$-acetone was added, the $^1$H NMR spectra of the resulting mixed sample were immediately measured and then measured continuously at intervals of approx. 2 min. The $^1$H NMR spectra of this reaction were measured for 30 min (Suppl. Fig. 29).

Etching [(C)(Au$^I$-TPP)$_6$](BF$_4$)$_2$ ($\mathbf{1^{TPP}}$) with $cis$-depe. A solution of $\mathbf{1^{TPP}}$ (2.0 ×10$^{-3}$ M) in $d_6$-acetone was prepared by dissolving $\mathbf{1^{TPP}}$ (2.9 mg, 1 μmol) in 0.5 mL $d_6$-acetone in the presence of an internal standard (IS, 1,3,5-trimethoxybenzene, 6 equiv., 6 μmol, 1.0 mg). We firstly measured the solution of $\mathbf{1^{TPP}}$ in $d_6$-acetone $^1$H NMR spectroscopy at 300 K. Then, once the solution of $cis$-depe (2.5 equiv., 3.3 ×10$^{-2}$ M, 75 μL) in $d_6$-acetone was added, the $^1$H NMR spectra of the resulting mixed sample were immediately measured and then measured continuously at intervals of approx. 2 min. The $^1$H NMR spectra of this reaction were measured for 18 min (Suppl. Fig. 30).

**Time-course experiments monitored by ESI-MS spectrometry.** Etching [(C)(Au$^I$-$SS$-NHC)$_6$](BF$_4$)$_2$ ($SS$-$\mathbf{1^{NHC}}$) with $cis$-depe. A solution of $SS$-$\mathbf{1^{NHC}}$ (5.0 × 10$^{-5}$ M) in acetone was prepared by dissolving $SS$-$\mathbf{1^{NHC}}$ (1.7 mg, 0.5 μmol) in 10 mL acetone. We first measured the original solution of $SS$-$\mathbf{1^{NHC}}$ (5.0 × 10$^{-5}$ M, 1 mL) in acetone by ESI-MS spectrometry. Then, once 10 μL of $cis$-depe (2 equiv., 0.01 M) in acetone was

added (meanwhile the solution was charged with a small magnetic stir and stirring), the ESI-MS spectra of the resulting reaction were measured as shown in Suppl. Fig. 31.

Etching $[(C)(Au^I\text{-}TPP)_6](BF_4)_2$ (**$1^{TPP}$**) with *cis*-depe. A solution of **$1^{TPP}$** ($5.0 \times 10^{-5}$ M) in acetone was prepared by dissolving **$1^{TPP}$** (1.5 mg, $5.0 \times 10^{-4}$ mmol) in acetone. We first measured the original solution of **$1^{TPP}$** ($5.0 \times 10^{-5}$ M, 1 mL) in acetone by ESI-MS spectrometry. Then, once 10 μL of *cis*-depe (2 equiv., 0.01 M) in acetone was added (meanwhile the solution was charged with a small magnetic stir and stirring), the ESI-MS spectra of the resulting reaction were measured. The intermediate species was observed as shown in Suppl. Fig. 32.

**Theoretical calculation details for etching mechanism.** The proposed intermediates including **Int1$^{NHC}$**, **Int2$^{NHC}$**, **Int1$^{TPP}$**, and **Int2$^{TPP}$** were simulated and optimised using the Gaussian 16 suite of programs[63]. Optimisation was performed using the B3LYP functional[59] combined with basis sets of 6-31G* (for C, N, P, H)[61] and LANL2DZ (for Au)[60]. The solvent effects were evaluated by single-point calculations in the optimised structure using the polarisable continuum model (PCM). All chemical species involved were optimised in the singlet state. The calculated energy profiles were illustrated in Suppl. Fig. 33.

## Data availability

The data that support the findings of this study are available from the corresponding authors upon request. The X-ray crystallographic data for the structures reported in this article have been deposited at the Cambridge Crystallographic Data Centre (CCDC) under deposition numbers CCDC 2280948 to CCDC 2280952. Source data are provided with this paper.

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

## Acknowledgements

This work was supported by JSPS KAKENHI Grant Numbers JP21H05022 (M.S.), JP22K14691 (X.P.), JP22H05133 (M.E.), MEXT KAKENHI Grant Number JP16H06509 (M.S.), and JST, CREST Grant Number JPMJCR22B2 (M.S.), Japan.

## Author contributions

M.S. and X.-L.P. designed the project, analysed the results, and prepared the manuscript. H.U. and Z.L. assisted in the synthetic experiments. P.Z. and M.E. performed the theoretical calculations and analyses. All authors were involved in revising the manuscript.

## Competing interests

The authors declare no competing interests.
