## [Peer Review File · Nature Communications]

Single-gold etching at the hypercarbon atom of C-centred hexagold(I) clusters protected by chiral N-heterocyclic carbenesReviewer #1 (Remarks to the Author):

ABSTRACT: Etching is an excellent top-down synthesis method for controlling the structures and chemical and physical properties of nanomaterials. Chemical etching at the atomic level is a particularly challenging task for the precise synthesis of nano-sized metal clusters. The synthesis of metal clusters containing smaller metal nuclei and potential surface vacancies will further elucidate the details of structure-function relationships and facilitate future materials design. Here we report the successful single-gold etching at a hypercarbon centre in gold(I) clusters. Specifically, C-centred hexagold(I) clusters protected by chiral N-heterocyclic carbenes were etched with bisphosphine to yield C-centred pentagold(I) (CAuI₅) clusters. The CAuI₅ clusters exhibit an unusually large bathochromic shift in luminescence, which is reproduced theoretically. The etching mechanism was experimentally and theoretically suggested to be a tandem dissociation-association-elimination pathway. Furthermore, the vacant site of the central carbon of the CAuI₅ cluster can accommodate AuCl, allowing for post-functionalisation of the C-centred gold(I) clusters.

Pei et al report the "Single-gold etching at the hypercarbon atom of C-centred hexagold(I) clusters protected by chiral N-heterocyclic carbenes"

The manuscript effectively presents a comprehensive overview of rationally conducted experiments including SCXRD and theoretical calculations to show the changes at the atomic level. Nevertheless, I have reservations regarding the paper's suitability for publication in Nature Communications. To enhance its compatibility with this prestigious journal, it would greatly benefit from a more thorough reflection on the novelty of the work making it align with that required for Nature Communications, as it is not very clear in the introduction section.

Authors have published work related to NHC-protected CAuI₆ clusters and in this work, they investigate whether etching occurs when bisphosphine is added to NHC-protected CAuI₆ clusters. While similar etching SS-1NHC and 1TPP with 1,2-bis(diphenylphosphino)benzene also yielded the corresponding CAuI₅ clusters (ref 26, and 46 in the manuscript).

Authors should include a chart showing all the clusters and their abbreviated labels to help readers navigate easily, and the literature compounds should be referenced.

The significance of the low flack parameter for SS, RR-2NHC should be mentioned briefly.

Reviewer #2 (Remarks to the Author):

The manuscript, "Single-gold etching at the hypercarbon atom of C-centred hexagold(I) clusters protected by chiral N-heterocyclic carbenes", is a comprehensive report on the photophysical properties of CAu(I)₅ cluster and mechanism of chemical etching method to reduce one Au(I) atom from CAu(I)₆ cluster. DFT calculations were performed to interpret the properties and to reinforce the proposed mechanism. This paper has a strong impact on how we can synthesize metal cluster by manipulating one-by-one chemical reaction. The unusual cluster-size dependence of the photophysical property is also intriguing. I think this manuscript is worth publishing in Nature Comm after minor comments are considered. No further review is necessary.

In abstract, 4th line, this sentence is hard to read. Please revise.

Is the same procedure applicable to form CAu(I)₄ cluster? The CAu(I)₄ cluster is charge neutral, and four Au(I) can fill the valency of the central C atom.

In the hexagonal and pentagonal clusters, the carbon atom should be in a very unusual electronic state. Please give a brief explanation.

The description about theoretical calculation is not enough and hard to reproduce. Please consider

reproducibility.

In the TDDFT calculation, B3LYP functional was used. This functional is not a long-range corrected one, and the description of charge-transfer type excitation should be carefully verified. In this case, TDDFT calculation reproduced the observed spectrum, and the performance of the functional was verified. But, I still feel that the authors give some comments on the B3LYP functional for application to the MLCT type excited states of the small Au clusters.

The unusual red-shift of the CAu(I)5 cluster from the CAu(I)6 cluster was explained on the basis of the HOMO-LUMO gap. The HOMO contains the C component, and the energy level shows significant high-energy shift. I would like the authors to give an explanation on the reason of the shift.

Figure 4f, Int2L should have one depe ligand, but the two depe ligands are eliminating in compound 3.

In Supplementary Fig 32, please give spin multiplicity of the system.

Reviewer #3 (Remarks to the Author):

The work is extremely well done and presented. This contribution should be well received by the Nature Comm readership. I have a few general questions:

1. Why are chiral NHCs used here - would achiral NHCs also work to demonstrate the generality of this surgical method?
2. if AuCl can be added as a cap to the Au octahedron could other metals be added as well? now this would open a new frontier in the field as luminescence properties could possibly be tuned in multimetallic systems. I am curious - could CuCl be added or other transition metal fragments?

Possibilities and questions abound. So an excellent contribution.

A few specific comments:

1. line 33: have made
2. line 43- are still a matter of debate
3. line 44 for surgically modifying metal clusters
4. line 51 - cannot be has made progress better: has attracted attention and significant advances have been made in this area.
5. in fig 2 show structure of depe

Reviewer #4 (Remarks to the Author):

Responses to Reviewer #1

ABSTRACT: Etching is an excellent top-down synthesis method for controlling the structures and chemical and physical properties of nanomaterials. Chemical etching at the atomic level is a particularly challenging task for the precise synthesis of nano-sized metal clusters. The synthesis of metal clusters containing smaller metal nuclei and potential surface vacancies will further elucidate the details of structure-function relationships and facilitate future materials design. Here we report the successful single-gold etching at a hypercarbon centre in gold(I) clusters. Specifically, C-centred hexagold(I) clusters protected by chiral N-heterocyclic carbenes were etched with bisphosphine to yield C-centred pentagold(I) (CAu₅) clusters. The CAu₅ clusters exhibit an unusually large bathochromic shift in luminescence, which is reproduced theoretically. The etching mechanism was experimentally and theoretically suggested to be a tandem dissociation-association-elimination pathway. Furthermore, the vacant site of the central carbon of the CAu₅ cluster can accommodate AuCl, allowing for post-functionalisation of the C-centred gold(I) clusters.

Pei et al report the “Single-gold etching at the hypercarbon atom of C-centred hexagold(I) clusters protected by chiral N-heterocyclic carbenes”

The manuscript effectively presents a comprehensive overview of rationally conducted experiments including SCXRD and theoretical calculations to show the changes at the atomic level. Nevertheless, I have reservations regarding the paper's suitability for publication in Nature Communications. To enhance its compatibility with this prestigious journal, it would greatly benefit from a more thorough reflection on the novelty of the work making it align with that required for Nature Communications, as it is not very clear in the introduction section.

Response: We thank you for your important comments that helped us improve the manuscript. In order to better emphasize the significance and novelty of the chemical etching method developed in this study, we have added a few statements to the "Introduction" and "Discussion" sections as shown below.

Revision in the manuscript

Page 3, "Introduction" section, paragraph 1

Before revision

Etching is an excellent top-down method to subtractively modify the structures and chemical and physical properties of a wide range of nanomaterials such as nanocrystals and colloidal nanoparticles¹⁻⁵, and nanoclusters⁶⁻¹¹ for a variety of applications (Fig. 1a).

(omission)

The ligand-exchange mechanisms of aggregative bimolecular nucleophilic substitution (S_N2)-like or unimolecular nucleophilic substitution (S_N1)-like type in nanogold regions containing Au^I and Au⁰ atoms are still a matter of debate²⁰⁻²². Despite the promise of chemical etching as a universal technique for precisely “surgical” metal clusters at the atomic level, little attention has been paid to ligand-protected Au^I clusters.

After revision

Etching is an excellent top-down method to **downsize** the structures **at the atomic level** and **modify the** chemical and physical properties of a wide range of nanomaterials such as nanocrystals and colloidal nanoparticles¹⁻⁵, and nanoclusters⁶⁻¹¹ for a variety of applications (Fig. 1a).

(omission)

It also remains controversial whether the ligand-exchange mechanisms **in nanogold regions containing Au^I and Au⁰ atoms is S_N2-like** bimolecular nucleophilic substitution or **S_N1-like type** unimolecular nucleophilic substitution²⁰⁻²². Despite the promise of chemical etching as a **general** technique **to downsize metal clusters** at the atomic level, little attention has been paid to ligand-protected Au^I clusters.

Page 3, "Introduction section", paragraph 2

Before revision

In particular, the hypercoordinated carbon (hypercarbon)³⁵-centred hexagold(I) (CAu^I₆) cluster [C(Au^I-L)₆]X₂ (L = ligand; X = counterion)²⁶ that bridges nano-sized metal clusters and organic molecules^{36,37} has recently made significant progress.

After revision

In particular, the hypercoordinated carbon (hypercarbon)³⁵-centred hexagold(I) (CAu^I₆) cluster [C(Au^I-L)₆]X₂ (L = ligand; X = counterion)²⁶ that bridges nano-sized metal clusters and organic molecules^{36,37} **has attracted attention and significant advances have been made in this area.**

Page 4, "Introduction" section, paragraph 3

Before revision

Here we have found that C-centred hexagold(I) clusters protected by chiral NHCs can be etched with a bisphosphine ligand to yield chiral NHC-protected CAu^I₅ clusters, and showed that the gold(I) clusters can be precisely operated at the atomic level by controlling the number of gold atoms centred at the hypercarbon (Fig. 1b). Furthermore, this etching method also works for the synthesis of TPP-protected CAu^I₅ analogues. Although there is a general rule that smaller gold clusters show more blue-shifted emission than larger clusters, the ligand-protected CAu^I₅ clusters show unusually red-shifted signals in both absorption and emission spectra compared to the CAu^I₆ counterparts, which was rationalised by theoretical calculations. Further experimental and theoretical studies suggest that the etching mechanism involves a tandem pathway of dissociation-association-elimination. Moreover, the reactivity of the CAu^I₅ clusters at the hypercarbon centre was further confirmed by the addition of AuCl.

After revision

Here, **we discovered** that **a chiral NHC-protected** C-centred hexagold(I) cluster can be etched with a bisphosphine ligand to **generate a** chiral NHC-protected CAu^I₅ **cluster. This was achieved by** controlling the number of gold atoms centred at the hypercarbon **at the atomic level** (Fig. 1b). Furthermore, this etching method **is also useful** for the synthesis of TPP-protected CAu^I₅ analogues. **In general,** smaller gold clusters show more blue-shifted emission than larger clusters, **but** the ligand-protected CAu^I₅ clusters show unusually red-shifted signals in both absorption and emission spectra compared to the CAu^I₆ counterparts, which was

rationalised by theoretical calculations. Further experimental and theoretical studies suggest that a tandem dissociation-association-elimination pathway is involved in the etching mechanism. The NHC-protected CAu^I₆ clusters are generally chemically stable, and the active site had to be placed on the hypercarbon to confer reactivity. In fact, CAu^I₅ synthesised by this method was highly reactive with AuCl, producing a CAu^I₆ cluster with a different ligand. Thus, the chemical etching method is expected to be an excellent way not only to reduce the size of metal ion clusters and significantly change their electronic structure, but also to asymmetrise the metal ion cluster structure and provide active sites.

Page 18, "Discussion" section, last sentence

Before revision

In particular, we can expect to gain new knowledge about reaction mechanisms such as addition reactions, elimination reactions, and catalytic reactions of organic compounds and their correlations.

After revision

Therefore, the chemical etching method is an excellent way to reduce the size of metal ion clusters and will be developed to control of the electronic structure, asymmetrisation of the metal ion cluster structure, catalytic reactions, and metal ion exchange.

Authors have published work related to NHC-protected CAu₁₆ clusters and in this work, they investigate whether etching occurs when bisphosphine is added to NHC-protected CAu₁₆ clusters. While similar etching SS-1NHC and 1TPP with 1,2-bis(diphenylphosphino)benzene also yielded the corresponding CAu₁₅ clusters(ref 26, and 46 in the manuscript).

Authors should include a chart showing all the clusters and their abbreviated labels to help readers navigate easily, and the literature compounds should be referenced.

Response: To make it easier for the readers to distinguish between all the clusters described in this paper, the compounds in the references are shown in Fig. 2 and the schematic diagram in Supplementary Fig.1.

Revision in the manuscript

Fig. 2, caption

Fig. 2 | Synthesis and characterisation of C-centred pentagold(I) clusters. **a** Etching syntheses of CAu^I₆ clusters^{26,43} to CAu^I₅ clusters (Supplementary Fig. 1) using (*cis*)-1,2-bis(diphenylphosphino)ethene (*cis*-depe), insets: photographs of crystals under a microscope under ambient light. **b** Single-crystal X-ray diffraction (SCXRD) structures of the cations [(C)(Au^I-L)₅]⁺ (L = *SS*- and *RR*-NHC) and the CAu^I₅ cores of *SS*- and *RR*-2^{NHC} with optically active *N*-heterocyclic carbene (NHC) ligands. **c** SCXRD structures of the cation [(C)(Au^I-TPP)₅]⁺⁴⁶. (omission)

Revision in the Supplementary Information

Supplementary Fig. 1. The schematic illustrations of the C-centered gold(I) clusters.

a, the CAu^I_6 clusters: $[(C)(Au^I-SS-NHC)_6](BF_4)_2$ ($SS-1^{NHC}$)⁹, $[(C)(Au^I-RR-NHC)_6](BF_4)_2$ ($RR-1^{NHC}$)⁹, $[(C)(Au^I-TPP)_6](BF_4)_2$ (1^{TPP})¹⁰. **b**, the CAu^I_5 clusters: $[(C)(Au^I-SS-NHC)_5](BF_4)$ ($SS-2^{NHC}$), $[(C)(Au^I-RR-NHC)_5](BF_4)$ ($RR-2^{NHC}$), $[(C)(Au^I-TPP)_5](BF_4)$ (2^{TPP})¹⁴.

Note: another $Au^I \cdots Au^I$ interaction was observed in the structure of 2^{TPP} , which was slightly different from the reported data¹⁴.

Supplementary References

14. Scherbaum, F., Grohmann, A., Müller, G. & Schmidbaur, H. Synthesis, structure, and bonding of the cation $[(C_6H_5)_3PAu]_5C^{\oplus}$. *Angew. Chem. Int. Ed. Engl.* **28**, 463–465 (1989).

The significance of the low flack parameter for SS, RR-2NHC should be mentioned briefly.

Response: Thanks for your important comments. We have added the following explanatory text for the low flack parameter for SS- and RR-2^{NHC}.

Revision in the manuscript

Page 8, "Results" section, paragraph 3

SS- and RR-2^{NHC} are crystallised in the I_4 space group with low flack parameters of 0.010(9) and $-0.027(11)$, respectively. Their flack parameters are very low (Supplementary Table 1), suggesting that optically pure molecules are packed.

Responses to Reviewer #2

The manuscript, “Single-gold etching at the hypercarbon atom of C-centred hexagold(I) clusters protected by chiral N-heterocyclic carbenes”, is a comprehensive report on the photophysical properties of CAu(I)₅ cluster and mechanism of chemical etching method to reduce one Au(I) atom from CAu(I)₆ cluster. DFT calculations were performed to interpret the properties and to reinforce the proposed mechanism. This paper has a strong impact on how we can synthesize metal cluster by manipulating one-by-one chemical reaction. The unusual cluster-size dependence of the photophysical property is also intriguing. I think this manuscript is worth publishing in Nature Comm after minor comments are considered. No further review is necessary.

Response: We thank the reviewer for the high evaluation of our study and for the valuable suggestions that helped us improve the manuscript. We have revised the manuscript according to the reviewer’s comments as follows.

In abstract, 4th line, this sentence is hard to read. Please revise.

Response: We thank you for the important comment. To improve the readability, the sentence on the 4th line in the manuscript as follows.

Revision in the abstract

Before revision

Etching is an excellent top-down synthesis method for controlling the structures and chemical and physical properties of nanomaterials. Chemical etching at the atomic level is a particularly challenging task for the precise synthesis of nano-sized metal clusters.

After revision

Chemical etching of nano-sized metal clusters at the atomic level has a high potential for creating metal number-specific structures and functions that are difficult to achieve with bottom-up synthesis methods. In particular, precisely etching metal atoms one by one from nonmetallic element-centred metal clusters and elucidating the relationship between their well-defined structures, and chemical and physical properties will open up new possibilities for metal clusters.

Is the same procedure applicable to form CAu(I)₄ cluster? The CAu(I)₄ cluster is charge neutral, and four Au(I) can fill the valency of the central C atom.

Response: Thank you for your very interesting remark. As the reviewer pointed out, the CAu₄ cluster has four C-Au^I bonds and is therefore charge neutral. A similar procedure could be applied to generate CAu₄ clusters, we would expect that an appropriate ligand would be needed to stabilise the CAu₄ cluster, as the Au^I...Au^I interactions could be significantly weaker. We hope to report the results of that study in the near future.

In the hexagonal and pentagonal clusters, the carbon atom should be in a very unusual

electronic state. Please give a brief explanation.

Response: The unique electronic structure of the central carbon atom is of interest. The orbital interactions have been discussed in previous studies^{50,51}. In this study, the bond orders of C-Au^I and Au^I...Au^I were analysed in detail as described in the main text (p. 9-10). For simplicity, the following sentence was added at the end of the 4th paragraph in the "Results" section.

Revision in the manuscript

Page 9, "Results" section, the end of paragraph 4

Before revision

(omission) Therefore, missing one Au atom in the CAu^I₅ cores may result in stronger C_{centre}-Au^I bonds, which may be important for stabilising the CAu^I₅ clusters.

After revision

(omission) Therefore, missing one Au atom in the CAu^I₅ cores may result in stronger C_{centre}-Au^I bonds, which may be important for stabilising the CAu^I₅ clusters. Moreover, the orbital interactions of the CAu^I₆ cluster were previously discussed in detail^{50,51}: the SS-1^{NHC} and SS-2^{NHC} clusters have [CAu₆]²⁺ and [CAu₅]⁺ cores, respectively, and the C-Au^I bond orders exhibit an unusual C-Au^I bond hypervalence.

References 50 and 51 are added:

50. Görling, A., Rösch, N., Ellis, D. E. & Schmidbaur, H. *Inorg. Chem.* **30**, 3986–3994 (1991).

51. Häberien, O. D., Schmidbaur, H. & Rösch, N. *J. Am. Chem. Soc.* **116**, 8241–8248 (1994).

The description about theoretical calculation is not enough and hard to reproduce. Please consider reproducibility.

Response: Details of the theoretical calculations are described in the "Theoretical calculation details for etching mechanism" section in the Supplementary Information on page S7 so that the results can be reproduced.

Revision in the Supplementary Information

Page S7, "Theoretical calculation details for etching mechanism" section

Before revision

The proposed intermediates including **Int1^{NHC}**, **Int2^{NHC}**, **Int1^{TPP}** and **Int2^{TPP}** were simulated and optimised using the Gaussian 16 suite of programs⁷. The calculated energy profiles were illustrated in Supplementary Fig. 32.

After revision

The proposed intermediates including **Int1^{NHC}**, **Int2^{NHC}**, **Int1^{TPP}** and **Int2^{TPP}** were simulated and optimised using the Gaussian 16 suite of programs⁷. Optimisation was performed using the B3LYP functional combined with basis sets of 6-31G* (for C, N, P, H) and LANL2DZ (for Au). The solvent effects were evaluated by single-point calculations in the optimised structure

using the polarisable continuum model (PCM). All chemical species involved were optimised in the singlet state. The calculated energy profiles were illustrated in Supplementary Fig. 33.

In the TDDFT calculation, B3LYP functional was used. This functional is not a long-range corrected one, and the description of charge-transfer type excitation should be carefully verified. In this case, TDDFT calculation reproduced the observed spectrum, and the performance of the functional was verified. But, I still feel that the authors give some comments on the B3LYP functional for application to the MLCT type excited states of the small Au clusters.

Response: We appreciate the reviewer for addressing this point. As reviewer noted, the MLCTs in the present small Au clusters do not show long-distance charge transfer and they essentially correspond to the charge reorganization and therefore, global hybrid functionals like B3LYP suitably describe these electronic transitions. We added this explanation in Supplementary Information (page S4).

Revision in the Supplementary Information

Page S4, "DFT and TD-DFT calculations" section

Before revision

We applied the B3LYP functional³ for geometry optimizations and TD-DFT calculations. The relativistic effective core potential LANL2DZ⁴ was used for the Au atoms, and the basis set for the other atoms was 6-31G*⁵. For simulating absorption spectra, (omission)

After revision

We applied the B3LYP functional³ for geometry optimisations and TD-DFT calculations. The relativistic effective core potential LANL2DZ⁴ was used for the Au atoms, and the basis set for the other atoms was 6-31G*⁵. Since MLCTs in the Au clusters etched in this study do not show long-distance charge transfer and essentially correspond to charge reorganisation, global hybrid functionals such as B3LYP adequately describe these electronic transitions. For simulating absorption spectra, (omission)

The unusual red-shift of the CAu(I)5 cluster from the CAu(I)6 cluster was explained on the basis of the HOMO-LUMO gap. The HOMO contains the C component, and the energy level shows significant high-energy shift. I would like the authors to give an explanation on the reason of the shift.

Response: The significant energy shift of the orbital energy levels is attributed to the total charge of the cluster. Indeed, the carbon component in the HOMO of the CAu^I₅ cluster (32.3%) is larger than that of the CAu^I₆ cluster (24.6%) (ref. 43, SI). The CAu^I₅ and CAu^I₆ clusters have total charges of +1 and +2, respectively, which leads to the large orbital energy difference or shifts between these clusters; the IPs (ionization potentials) of C and Au are 11.2 and 9.2 eV, respectively. Considering the orbital interactions, the ratio of carbon in the HOMO cannot explain the significant energy shift.

Figure 4f, Int2L should have one depe ligand, but the two depe ligands are eliminating in compound 3.

Response: The **Int2^L** (L = NHC, TPP) shown in **Fig. 4f** has only one *cis*-depe ligand, while the product of compound **3** [Au(*cis*-depe)₂(BF₄)] has two *cis*-depe ligands. This is because a slight excess of *cis*-depe (2.5 equiv) was added to the dichloromethane solution of the CAu^I₆ cluster (1 equiv). After removal of [*cis*-depeAu]⁺ in **Int2^L** (L = NHC, TPP), the remaining *cis*-depe in solution would bind with [*cis*-depeAu]⁺ to form the stable species of compound **3**. To clarify this point, **Fig. 4f** was modified as follows.

Revision in the Figure

Fig. 4| f A proposed etching mechanism with two intermediates **Int1^L** and **Int2^L** (L = NHC, TPP), with **Int1^{TPP}** and **Int2^{TPP}** detected by ESI-MS spectrometry (Supplementary Fig. 32). Colour code: Au, yellow; C, grey; L, cyan; P, orange.

In Supplementary Fig 32, please give spin multiplicity of the system.

Response: The spin states of the CAu^I₅ and CAu^I₆ clusters are singlet states. This is discussed in the caption of this figure.

Responses to Reviewer #3

The work is extremely well done and presented. This contribution should be well received by the Nature Comm readership.

Response: We appreciate the reviewer's positive comments.

I have a few general questions:

1. Why are chiral NHCs used here - would achiral NHCs also work to demonstrate the generality of this surgical method?

Response: We would like to thank you for your very important question. The chiral NHCs used in this study were found to be useful ligands that highly stabilise asymmetrically twisted optically pure CAu^{I}_6 clusters (see our previous paper: *J. Am. Chem. Soc.* **144**, 2156–2163 (2022)). As part of our systematic study of ligand effects, we discovered that the addition of bis-phosphine caused a unique etching reaction on the chiral NHCs-protected CAu^{I}_6 clusters. We have previously reported some achiral NHC-protected CAu^{I}_6 clusters, including $\text{BliPr-CAu}^{\text{I}}_6$ ($\text{BliPr} = N,N'$ -diisopropylbenzimidazolylidene, *Bull. Chem. Soc. Jpn.* **94**, 1324–1330 (2021)), the corresponding $\text{BliPr-CAu}^{\text{I}}_5$ was observed by ESI-MS, but crystallisation of the CAu^{I}_5 cluster preferentially yielded the corresponding CAu^{I}_6 cluster, probably due to the high reactivity of the CAu^{I}_5 cluster with the BliPr ligand. Therefore, it is worth continuing to investigate the generality of our method.

2. if AuCl can be added as a cap to the Au octahedron could other metals be added as well? now this would open a new frontier in the field as luminescence properties could possibly be tuned in multimetallic systems. I am curious - could CuCl be added or other transition metal fragments?

Response: We would like to thank you for a very interesting point. Adding other metals as caps to form new octahedral multi-metallic clusters is certainly promising. Related research is currently under investigation in our laboratory. Although there are no reports we have successfully used a silver(I) salt as a cap. Other trials by adding CuCl and other transition metal fragments are still in progress.

Possibilities and questions abound. So an excellent contribution.

A few specific comments:

1. line 33: have made ...
2. line 43- are still a matter of debate
3. line 44 for surgically modifying metal clusters
4. line 51 - cannot be has made progress better: has attracted attention and significant advances have been made in this area.
5. in fig 2 show structure of depe

Response: We would like to thank you for your very positive comments. Our response to each comment is as follows.

Revision in the manuscript

To comment 1. line 33: have made ...

Page 3, "Introduction" section, paragraph 1

Revised according to the reviewer's suggestion.

Before revision

For example, chemical etching methods that involve ligand engineering made great advances,

After revision

For example, chemical etching methods that involve ligand engineering **have** made great advances,

To comment 2. line 43- are still a matter of debate

Page 3, "Introduction" section, paragraph 1

In order to better emphasize the significance and novelty of the chemical etching method developed in this study, we have added a few statements to the "Introduction" section as shown below.

Before revision

Etching is an excellent top-down method to subtractively modify the structures and chemical and physical properties of a wide range of nanomaterials such as nanocrystals and colloidal nanoparticles¹⁻⁵, and nanoclusters⁶⁻¹¹ for a variety of applications (Fig. 1a).

(omission)

The ligand-exchange mechanisms of aggregative bimolecular nucleophilic substitution (S_N2)-like or unimolecular nucleophilic substitution (S_N1)-like type in nanogold regions containing Au^I and Au^0 atoms are still a matter of debate²⁰⁻²². Despite the promise of chemical etching as a universal technique for precisely "surgical" metal clusters at the atomic level, little attention has been paid to ligand-protected Au^I clusters.

After revision

Etching is an excellent top-down method to **downsize** the structures **at the atomic level** and **modify the** chemical and physical properties of a wide range of nanomaterials such as nanocrystals and colloidal nanoparticles¹⁻⁵, and nanoclusters⁶⁻¹¹ for a variety of applications (Fig. 1a).

(omission)

It also remains controversial whether the ligand-exchange mechanisms **in nanogold regions containing Au^I and Au^0 atoms is S_N2 -like** bimolecular nucleophilic substitution or **S_N1 -like type unimolecular nucleophilic substitution**²⁰⁻²². Despite the promise of chemical etching as a **general** technique **to downsize metal clusters** at the atomic level, little attention has been paid to ligand-protected Au^I clusters.

To comment 3. line 44 for surgically modifying metal clusters

Please see our response to your comment 2.

To comment 4. line 51 - cannot be has made progress better: has attracted attention and significant advances have been made in this area.

Revised according to the reviewer's suggestion.

To comment 5. in fig 2 show structure of depe

Fig. 2 | Synthesis and characterisation of C-centred pentagold(I) clusters. **a** Etching syntheses of CAu^I₆ clusters^{26,43} to CAu^I₅ clusters (Supplementary Fig. 1) using (*cis*)-1,2-bis(diphenylphosphino)ethene (*cis*-depe), insets: photographs of crystals under a microscope under ambient light. **b** Single-crystal X-ray diffraction (SCXRD) structures of the cations [(C)(Au^I-L)₅]⁺ (L = SS- and RR-NHC) and the CAu^I₅ cores of SS- and RR-2^{NHC} with optically active *N*-heterocyclic carbene (NHC) ligands. **c** SCXRD structures of the cation [(C)(Au^I-TPP)₅]⁺⁴⁶ ...

Responses to Reviewer #4

Response: We thank you for taking your precious time to review our manuscript.